# Total replacement of fishmeal with poultry by-product meal affected the growth, muscle quality, histological structure, antioxidant capacity and immune response of juvenile barramundi, *Lates calcarifer*

**Md. Reaz Chaklader**[1]*, **Muhammad A. B. Siddik**[2], **Ravi Fotedar**[1]

1 School of Molecular and Life Sciences, Curtin University, Bentley, WA, Australia, 2 Department of Fisheries Biology & Genetics, Faculty of Fisheries, Patuakhali Science and Technology University, Patuakhali, Bangladesh

* mrreaz.chaklader@postgrad.curtin.edu.au

**Data Availability Statement:** All data generated or analyzed during this study have been presented in the forms of figures and tables in the paper.

## Abstract

The present study investigates if the total replacement of dietary fishmeal (FM) with poultry by-product meal (PBM), supplemented with methionine influences the muscle fatty acids composition, normal gut morphology, histological traits of the liver, muscle, and gill, liver enzymes, immune and antioxidant response, and stress-related gene in juvenile barramundi, *Lates calcarifer* in relation to growth and feed utilization. Barramundi (3.58±0.01g) were randomly distributed into six 300 L seawater recirculating tanks (25 fish/tank) and fed two formulated isonitrogenous and isolipidic diets for 6 weeks. The control diet had FM as the sole animal protein source, whereas other test diet had only PBM as an animal protein source. Dietary PBM affected the fish performance and feed utilization. Regarding muscle fatty acid profile, total saturated fatty acids and monounsaturated fatty acids elevated while total PUFA particularly n-3 LC-PUFA and EPA decreased in PBM fed fish than control diet fed fish. Liver, muscle, gill, and intestinal histology showed no obvious alteration in control diet fed fish, however, more lipid droplets and hepatic vacuolization in the liver, necrotic myotome in muscle, hyperplasia in secondary lamellae in gill and short and broken folds in the intestine were observed in PBM fed fish. Similar to light microscopy observation of intestinal morphology, the transmission electron microscopy (TEM) analysis revealed shorter and smaller microvilli in fish fed PBM. Histopathological alterations in the liver of PBM fed fish were further associated with the elevated levels of aspartate aminotransferase (AST) and glutamate dehydrogenase (GLDH) and the significant upregulation of stress-related genes, HSP70 and HSP90. Also, a negative influence on lysozyme activity, and antioxidant enzymatic activities were recorded in fish fed PBM. Overall, it can be concluded that a total substitution of FM protein by methionine supplemented PBM negatively influenced the growth performance, liver health, histological traits of different organs, immune and antioxidant response, and expression of stress-related genes in juvenile barramundi.

**Funding:** Support of the trial was obtained from the Research Training Program (RTP) Stipend Scholarship, funded by Australian Government to Md Reaz Chaklader (No. 19061054-Curtin).

**Competing interests:** The authors declare no competing interests.

## Introduction

One of the major bottlenecks for carnivorous aquafeed production is the inconsistence supply of global fishmeal (FM) and escalated prices. Therefore, efforts have been exerted over several decades to investigate the feasibility of alternative dietary protein sources replacing FM in carnivore finfish aquaculture [1–4]. Presently, three main categories of FM replacements including terrestrial plant meals, rendered animal by-products, and seafood processing wastes are commercially available and used [5]. However, more than 50% of substitution of FM is now regularly achieved commercially in most carnivorous species [5], including barramundi or Asian sea bass, *Lates calcarifer* a commercial important carnivorous fish species [6]. Barramundi has good meat quality, ability to tolerate a wide range of salinity, and ability to adapt to the versatile farming environment [7]. It is popularly cultivated both in freshwater and seawater in Malaysia, Thailand, Taiwan, Indonesia, Saudi Arabia, and Australia, contributing USD 320 million globally [8, 9]. In Australia, barramundi farming is heavily dependent on imported FM resulting in incurring around 40% diet related cost which is the main impediment to increase the profitability [10]. Hence, nutritional studies on barramundi have commenced since the 1980s [11] and many of the studies have dedicated to replacing the FM with rendered animal meals [7, 10, 12–18] or plant meals [12, 13, 19–24].

Poultry by-product meal (PBM), an economical and easily available ingredient compared to FM contains a higher level of protein and most of the indispensable amino acids except for lysine and methionine [25–27]. Although, significant research in Australia and New Zealand has been conducted to commercially utilize PBM in various industries, its worldwide utilization is controlled by several regulations, for example the ban in European Union that has been recently lifted to allow the utilization of non-ruminant processed animal protein for aquaculture species [5]. Like other animal-based protein, another major limitation regarding the utilization of PBM is the variable digestibility due to variability in its composition and quality [28]. There have been several studies examining the effect of PBM on barramundi but results are mixed. For instance, Glencross, Blyth [29] reported that the inclusion of poultry meals up to 338g/kg does not influence the growth but beyond this level had a deleterious effect. Besides, a recent study of Simon, Salini [8] found that barramundi growth was impaired despite feeding balanced poultry protein concentrate (5–20%) while PBM along with supplementation of tuna hydrolysate could replace 90% of FM without impairing the growth [16]. In our earlier study, regardless of full-fat black soldier fly larvae supplementation, barramundi fed 90% PBM impacted the growth performance [18]. Similarly, the utilization of PBM above 50% affected the welfare of some marine fish species [25, 26, 30–32]. Imbalanced dietary essential amino acid (EAA) particularly methionine and lysine in PBM based diets are one of the major causes resulting growth depression in many fish [28]. The methionine requirement for barramundi was reported to be 2.2% [11]. In this study, methionine was supplemented to PBM to investigate if supplementation of deficient EAA in PBM could substitute FM totally.

In addition to assessing the growth performance related to non-FM protein ingredients, health aspects including the changes in serum biochemical assays, immune responses, and stress-related oxidative biomarkers are also crucial parameters of interest to aquaculturists. Nutritional factors can influence the production of oxidative enzymes [33]. Oxidative stress is characterized by an increase in malondialdehyde (MDA) and a decrease in glutathione peroxidase (GPx) [34]. The imbalance between antioxidant defences and free radical generation may cause cell damage, which may provoke the leaking of liver enzymes particularly ALT and GLDH in fish [35], associated with liver cell damage. Although published data are available on the effects of plant protein on oxidative biomarkers of barramundi [21, 22, 36, 37], there is less information on the effects of animal protein inclusions. The dietary inclusion of PBM

impacted the liver health of barramundi by increasing the levels of ALT and GLDH [18], and animal protein ingredients elevated the levels of AST and ALT in hybrid grouper, *Epinephelus fuscoguttatus♀× Epinephelus lanceolatus♂* [38]. Moreover, the equilibrium between oxidants and antioxidants is also important for immune cell function since it preserves the integrity and functionality of the cell membrane. Hence, it is crucial to understand the correlation between the antioxidants, liver enzymes, and immune response when entire dietary FM is replaced by any alternate- protein ingredients.

During dietary modification, it is important to consider that replacement of FM with potential ingredients do not exert adverse effects on the welfare of the tested species, as the welfare of fish in captive condition has been a growing concern over the decades [39–41]. The histological approach is one of the important frontline tools applied to assess the health status that can be achieved by evaluating the morphological status of different organs. The liver is the biggest organs involved in nutrient metabolism and producing biochemical compounds required for digestion. Muscle structure is also important as it reflects the nutritional condition and agility of the fish. The intestine is a primary immune organ in fish participated in digestion and absorption of nutrients as well as defence mechanism against microbes [42]. The evaluation of histopathological changes in these organs is important to assess the non-FM diet. Therefore, the present study aimed to investigate whether methionine supplemented PBM based diet has an ability to substitute FM completely without compromising growth performance, fatty acids composition, histological traits of different organs, serum biochemical response, stress-related genes expression, and antioxidant activities in barramundi.

## Materials and methods

### Animal ethical statement

The experiment was conducted at Curtin Aquatic Research Laboratory (CARL) in Curtin University, Australia in compliance with relevant guidelines and regulations set by the Australian Code of Practice for the care and use of animals for scientific purposes. All methods involving fish were reviewed and approved by the Curtin University Animal Ethics Committee (ARE2018-37). Prior to handling fish, AQUI-S® was used as anaesthesia and an overdose of AQUI-S was used as euthanasia to minimise stress, pain, and discomfort to the fish following the protocol of the Curtin Research Laboratories standard operating procedure (SOP) of anaesthetizing and euthanizing of fish.

### Experimental diets

Except PBM, all the ingredients required for formulating test diets were purchased from the Special Feeds, 3150 Great Eastern Hwy, Glen Forrest, WA. Two isonitrogenous and isolipidic containing approximately 48% crude protein and 13% crude lipid were prepared to meet the nutritional requirement of barramundi [43]. FM and PBM were used as the main protein source and canola oil and cod liver oil were used as lipid sources. A control diet was prepared based on FM and another diet was formulated by replacing 100% of FM with PBM supplemented with 0.40% methionine (Table 1) to meet the established methionine requirement for normal growth of barramundi [11, 44]. The diets were formulated in compliance with the standard protocol of CARL. Briefly, all the dry ingredients were mixed homogeneously using a food mixture (Hobart Food equipment, Australia) before blending with fish oil and distilled warm water to make a stiff dough. The dough was passed through a mincer to make 3 mm pellets, then spread out and dried in an oven at 60˚C for 36 hours. After drying, pellets were sealed in plastic bags before refrigerating at 4˚C until used in the feeding trial. The fatty acid and amino acid profile of test diets and PBM is shown in Table 2 and Table 3, respectively.

**Table 1. Formulation and proximate composition of test diets for barramundi.**

| Ingredients[a] (g/100g DM) | Control | 100PBM |
|---|---|---|
| †FM | 72.00 | 0.00 |
| ‡PBM[b] | 0.00 | 69.50 |
| Canola oil | 1.00 | 3.00 |
| Cod liver oil | 0.50 | 6.00 |
| Corn/wheat starch | 7.00 | 7.00 |
| wheat (10 CP) | 16.90 | 11.50 |
| Lecithin—Soy (70%) | 1.00 | 1.00 |
| Vitamin C | 0.05 | 0.05 |
| Dicalcium Phosphate | 0.05 | 0.05 |
| Methionine | 0.00 | 0.40 |
| Vitamin and mineral premix | 0.50 | 0.50 |
| Salt (NaCl) | 1.00 | 1.00 |
| *Proximate composition (% dry weight)[c]* | | |
| Moisture | 14.96 | 13.98 |
| Crude Protein | 47.88 | 47.86 |
| Crude Lipid | 12.59 | 12.71 |
| Ash | 9.67 | 10.24 |
| Gross energy (MJ/kg) | 20.23 | 19.95 |

[a] Specialty Feeds, Glen Forrest Stockfeeders, 3150 Great Eastern Highway, Glen Forrest, Western Australia 6071.

[b] Kindly provided by Derby Industries Pty Ltd T/A, Talloman Lot Lakes Rd, Hazelmere WA 6055.

[c] Analysed according to Association of Official Analytical Chemists (AOAC) [45].

† FM (Fishmeal): 64.0% crude protein, 10.76% crude lipid and 19.12% ash.

‡ PBM (Poultry by-product meal): 67.13% crude protein, 13.52% crude lipid and 13.34% ash.

## Fish husbandry and management

Three hundred and fifty barramundi were obtained from the Australian Centre for Applied Aquaculture Research (ACAAR), Fremantle, Australia in oxygenated plastic bags. Prior to commencing the trial, all fish were stocked into two fiberglass tanks (300 L) filled with ocean water and fed a commercial diet (470 g protein kg$^{-1}$ diet and 20.0 MJ kg$^{-1}$dietary gross energy) twice daily for two weeks to acclimate them to CARL experimental facilities and conditions. Following acclimation, 150 normal and visually healthy fish averaging (3.58±0.01g) were randomly distributed into six 300-L tanks, containing 250 L water in each tank. Therefore, stocking number of barramundi in each tank was 25. Each tank was equipped with an aerator, electric heater, and external bio-filter (Astro® 2212, China) to maintain DO, temperature, and other water quality parameters at an optimal level. Hence, the temperature was maintained at 27.90–29.20˚C, dissolved oxygen (DO) at 5.92–7.42 mgL$^{-1}$, salinity at 32–36 ppt, and photoperiod as 14:10 h LD. Commercial test kits were used to test ammonia nitrogen (<0.50 mgL$^{-1}$) and nitrite (<0.50 mgL$^{-1}$) level regularly. Each test diet had three replicates and fed by hand twice daily at 8.00 am and 6.00 pm to visual satiety levels for 42 days. Uneaten feed, if any, was collected by siphoning to calculate feed intake, and the number of dead fish were monitored daily to assess the fish survival rate. After 42 days, all fish were starved for 24 h prior to weighing total biomass to analyse the growth performance.

## Fatty acids profile

Fish muscles in the form of three samples per dietary treatment were used for fatty acids analysis. Four fish muscle was filleted, wrapped with aluminium foil, freeze-dried, and pooled together.

**Table 2. Fatty acids (mg/100g of dry sample) composition of control and test diet replacing FM totally with PBM in barramundi.**

| Fatty acids | Experimental diets | | |
| --- | --- | --- | --- |
| | Control | 100PBM | PBM |
| C12:0 | 2.73 | 7.11 | 9.39 |
| C14:0 | 131.63 | 342.45 | 73.69 |
| C16:0 | 1161.21 | 2090.88 | 2336.27 |
| ΣSFA[1] | 1981.40 | 3216.29 | 3344.73 |
| C14:1n5 | 1.52 | 11.32 | 16.26 |
| C16:1n7 | 165.22 | 435.58 | 540.27 |
| C18:1cis+trans | 1158.94 | 3800.14 | 4410.64 |
| C20:1 | 79.86 | 483.71 | 60.84 |
| ΣMUFA[2] | 1482.21 | 4873.58 | 5057.95 |
| C18:3n3 | 120.20 | 285.67 | 260.39 |
| C20:5n3 (EPA) | 178.50 | 278.99 | 16.79 |
| C22:5n3# | 63.30 | 64.60 | 36.67 |
| C22:6n3 (DHA) | 908.53 | 455.23 | 27.47 |
| Σn-3 PUFA[3] | 1309.12 | 1240.95 | 358.96 |
| C18:3n6 | 8.84 | 10.58 | 23.07 |
| C20:3n6 | 15.50 | 18.76 | 56.76 |
| C20:4n6 | 112.83 | 43.18 | 180.13 |
| C22:4n6# | 91.14 | 16.39 | 4.78 |
| Σn-6 PUFA | 228.31 | 88.91 | 264.74 |
| Σn-3/n-6 | 5.73 | 13.96 | 1.36 |
| ΣPUFA[4] | 2184.01 | 2437.48 | 2386.97 |
| Σn-3 LC-PUFA | 1158.61 | 806.57 | 85.36 |

[1]Contains 10:0, 13:0, 15:0, 17:0, 18:0, 20:0, 21:0, 22:0 and 23:0.

[2]Contains C15:1, C17:1, C22:1n9 and C24:1.

[3]Contains C18:4n3, C20:3n3.

[4]Contains C18:2 trans, C18:2 cis, C20:2, C22:2.

Poultry by-product meal, PBM; saturated fatty acids, SFA; monounsaturated fatty acids, MUFA; polyunsaturated fatty acids, PUFA.

Eicosapentaenoic acid, EPA; DHA, docosahexaenoic acid, sum of saturated fatty acids, ΣSFA; sum of monounsaturated fatty acids, ΣMUFA; sum of polyunsaturated fatty acids, ΣPUFA; sum of omega-3 polyunsaturated fatty acids, Σn-3 PUFA; sum of omega-6 polyunsaturated fatty acids, Σn-6 PUFA and LC-PUFA, long-chain polyunsaturated fatty acids (sum of 20:3n-3, 20:5n-3, 22:5n-3 and 22:6n-3).

The fatty acids profile of experimental diets and fish flesh was carried out following the protocol of O'Fallon, Busboom [47], and Siddik, Chungu [7]. Approximately 0.5g of sample was hydrolysed at 55°C for 1.5 hrs with 0.1ml of internal standard (1.2g nonadecanoic acid in 100ml chloroform), 0.7ml of 10N KOH and 5.3ml of methanol. The sample was then methylated at 55°C for 1.5hrs with 0.6mL of 24N of sulphuric acid. The FAMES was extracted into 1ml of hexane and then quantified gas chromatography with flame ionization detection. The column used was a capillary column HP INNOWax GC column (60m x 0.25mm ID film 0.50 micron) with hydrogen as the carrier gas. Each sample were run in triplicate and results are expressed as an average.

## Histological and Transmission Electron Micrograph (TEM) analysis

After 42 days of feeding, one fish from each tank was randomly euthanized with AQUI-S at 175 mg/L to excise liver, muscle, gill, and intestine for histological and TEM evaluation in

**Table 3. Amino acids (g/100g on dry matter basis) composition of test diets and PBM.**

| | Experimental diets | | |
|---|---|---|---|
| *Amino acids*[a] | Control | 100PBM | PBM |
| Hydroxyproline | 1.7 | 3.2 | 3.2 |
| Histidine | 2.4 | 1.8 | 1.8 |
| Taurine | 0.5 | 0.5 | 0.4 |
| Serine | 5.3 | 5.0 | 5.0 |
| Arginine | 4.5 | 4.8 | 5.1 |
| Glycine | 13.2 | 16.4 | 16.5 |
| Aspartic acid | 8.8 | 7.8 | 7.8 |
| Glutamic acid | 11.7 | 12.1 | 11.5 |
| Threonine | 4.9 | 4.2 | 4.3 |
| Alanine | 9.4 | 9.1 | 9.1 |
| Proline | 6.1 | 7.3 | 7.0 |
| Lysine | 6.2 | 5.3 | 5.5 |
| Tyrosine | 2.0 | 1.8 | 2.0 |
| Methionine | 2.4 | 2.2 | 1.8 |
| Valine | 5.6 | 5.2 | 5.1 |
| Isoleucine | 4.3 | 3.8 | 3.8 |
| Leucine | 7.6 | 6.4 | 6.9 |
| Phenylalanine | 3.3 | 3.0 | 3.0 |

[a]Determined including hydroxyproline and taurine analysis following our earlier study [46].

response to test diets. For histological analysis, samples of all organs were fixed immediately in 10% buffered formalin, subsequently dehydrated with series of ethanol, infiltrated in xylene, and embedded in paraffin wax, as per standard histological protocols. Section of approximately 5 μm thickness was stained with Periodic Acid-Schiff (PAS) and digitally photographed under a light microscope (BX40F4, Olympus, Tokyo, Japan).

For TEM analysis, freshly collected intestinal samples washed in 2.5% glutaraldehyde buffered in 1x PBS at pH 7.4 before performing secondary fixation in 1% OsO4 (80 W 2 min on, 2 min off, 2 min on), dehydrating in ethanol (50, 70, 95 and 100% at 250 W, 40 seach) and infiltrating finally with epoxy resin in acetone (Procure 812, Proscitech) (1:3, 1:1, 3:1ratios at 250 W, 3 min each). Samples were processed as described in the earlier study in our lab [16] and screened a LaB6 TEM (JEOL2100, Japan) at 120 kV. The electron micrographs obtained from TEM analysis at 30,000 magnification were analysed using ImageJ (National Institute of Health, USA) to determine microvilli length and diameter.

## Antioxidant status assessment

The enzyme activities of serum malondialdehyde (MDA) was determined using commercial assay kits following the manufacturer's instructions (Bockit, BIOQUOCHEM SL, 33428 Llanera-Asturias, Spain) and glutathione peroxidase (GPx) was measured with the Randox Laboratories test combination (Ransel, Antrim, United Kingdom) following the protocol of earlier study in our laboratory [22].

## Serum biochemistry and immunity

Fish were captured gently at 42 days post-feeding, immediately dipped in a bucket containing 8 mg l$^{-1}$ of AQUI-S$^{®}$, and blood was taken by puncturing caudal vessels using 1 mL non-

**Table 4. Primers of qPCR used in the experiment.**

| Genes | Sequences (5′ - 3′) | | Product size | Tm (˚C) | |
|---|---|---|---|---|---|
| Heat shock protein kDa70, HSP70 | F: AAGGCAGAGGATGATGTC | | 186 | 59 | Mohd-Shaharuddin, Mohd-Adnan [48] |
| | R: TGCAGTCTGGTTCTTGTC | | | | |
| Heat shock protein kDa90, HSP90 | F: ACCTCCCTCACAGAATACC | | 197 | 59 | Mohd-Shaharuddin, Mohd-Adnan [48] |
| | R: CTCTTGCCATCAAACTCC | | | | |
| 18S rRNA, 18S | F: TGGTTAATTCCGATAACGAACGA | | 94 | 59/60 | Mohd-Shaharuddin, Mohd-Adnan [48] |
| | R: CGCCACTTGTCCCTCTAAGAA | | | | |
| Elongation factor-1α, ef1α | F: AAATTGGCGGTATTGGAAC | | 83 | 59/60 | Mohd-Shaharuddin, Mohd-Adnan [48] |
| | R: GGGAGCAAAGGTGACGAC | | | | |

heparinized syringes (22G). Blood was allowed to clot for 24 h at 4˚C, centrifuged for 15 min at 3000 rpm and 4˚C, the serum collected and stored immediately at—80˚C for the analysis of serum biochemical parameters, oxidative biomarkers, and immune parameters. Serum clinical chemistry and immune-related parameters were analysed according to the protocol of our earlier study [18, 46].

## RNA extraction and qRT-PCR analysis

Liver from control and PBM fed fish were aseptically collected after euthanizing (AQUI-S, 175 mg l$^{-1}$) the fish and preserved in RNA Later (Sigma-Aldrich, Germany) at—80˚C until RNA extraction. Five milligrams of liver tissue stored in RNA Later was used for RNA extraction using RNeasy Mini Kit (Qiagen, Hilden, Germany) according to manufacturer protocol. The quality of RNA was checked by gel electrophoresis and, the purity and quantity were determined gel electrophoresis before synthesizing complementary DNA (cDNA) from 1 μg of total RNA using Omnicript RT kit (Qiagen, Hilden, Germany) following the instruction of manufacturer's company. qRT-PCR on stress-related genes were performed by PowerUp$^{TM}$ Cyber Green Master Mix (Thermo Scientific, USA) with 7500 Real-Time PCR System (Applied Biosystems, USA) and data were normalised against housekeeping genes, *18S rRNA* and *Ef1-a*, (Table 4) and analysed using REST$^{©}$ software [48].

## Calculation and statistics

Specific growth rate (SGR), feed conversion ratio (FCR) and total feed intake (TFI) were calculated using the following equations-

$$\text{Weight gain (WG, g)} = [(\text{Mean final weight} - \text{Mean initial weight})/(\text{Mean initial weight})]$$

$$\begin{aligned}\text{Specific growth rate (SGR, \%d)} \\= [(\ln(\text{final body weight}) - \ln(\text{pooled initial weight}))/\text{Days}] \times 100\end{aligned}$$

$$\text{Feed conversion ratio (FCR)} = [(\text{dry feed fed})/(\text{wet weigth gain})]$$

$$\begin{aligned}\text{Feed intake (FI, g/fish d}^{-1}) \\= [(\text{Dry diet given} - \text{Dry remaining diet recovered})/\text{days of experiment})/\text{ no.of fish}]\end{aligned}$$

All data were represented as mean±SE. The differences between control and PBM fed fish in all data were determined by unpaired student *t*-test at the significance level of $0.05 < P < 0.001$. Percent survival at the termination of the feeding trial was plotted using the Kaplan-Meier survival method with the Log-rank (Mantel-Cox) test.

# Results

## Growth performance, feed utilization and survival

Fish growth, feed intake, and survival rate in response to 42 days feeding trial are presented in Table 5 and Fig 1. The mean final body weight (FBW) and specific growth rate (SGR) of fish fed PBM were significantly lower than the FBG and SGR of fish fed the control diet. FCR in PBM fed fish increased with lower feed intake in PBM fed fish. Survival rate (Fig 1), as drawn by Kaplan-Meier survival analysis with 95% confidence at the end of the 42 days trial decreased significantly in PBM fed fish (81.33%) than the control (93.33%) ($\chi^2_{100PBM}$ = 4.514, df = 1, $P$ = 0.034).

## Muscle fatty acids composition

The FAs profile of barramundi muscle at the termination of 42 days trial was influenced by the PBM based diet (Table 6). The dietary inclusion of PBM significantly augmented total SFA. All SFA including capric acid (C10:0), lauric acid (C12:0), myristic acid (C14:0), palmitic acid (C16:0), margaric acid (C17:0), stearic acid (C18:0), arachidic acid (C20:0), heneicosylic acid (C21:0), behenic acid (C22:0) and tricosylic acid (C23:0) except for tridecylic acid (C13:0), and pentadecylic acid (C15:0) were significantly higher in the muscles of fish fed PBM. Similarly, total MUFA concentration increased in the muscle of PBM fed fish. PUFA differed significantly between the test diets, with lower concentration of n-3 LC-PUFA and C22:4n6 in PBM fed fish than the fish fed the control diet. Similar result was recorded in ∑n-3/∑n-6 ratio.

## Histopathology of liver, muscle, gill and intestine

Total replacement of FM with PBM dysregulated the histological structure of liver, muscle, gills, and intestine (Fig 2A–2H). The liver of control (Fig 2A) fed fish showed higher pigmentation of hepatocyte cytoplasm, indicating a higher amount of glycogen, while the liver of PBM fed fish (Fig 2B) showed less hepatocyte cytoplasm pigmentation, indicating less amount of glycogen with more lipid vacuolization. Healthy and normal myotome were observed in the muscle of the fish fed the control diet (Fig 2C) but necrotic myotome was found in the fish fed PBM diet (Fig 2D). Control fed fish showed normal gill structure (Fig 2E) but hyperplasia in secondary lamellae was recorded in PBM fed fish (Fig 2F). All the examined fish fed control (Fig 2G) presented normal intestinal structure whilst broken and short fold were found in fish fed PBM (Fig 2H).

**Table 5. Fish performance including Final Body Weight (FBW), Weight Gain (WG), Specific Growth Rate (SGR), Feed Intake (FI), and Feed Conversion Ratio (FCR) of barramundi when fed control and PBM based diet over a period of 42 days.**

| Growth performance | Experimental diets | | Unpaired t-test | |
|---|---|---|---|---|
| | Control | 100PBM | *t*-value | *P*-value |
| IW (g) | 3.52±0.02 | 3.49±0.06 | 0.82 | 0.46 |
| FBW (g) | 54.91±0.55[a] | 32.67±0.23[b] | 37.38 | 0.00 |
| WG (g) | 51.39±0.53[a] | 29.18±0.23[b] | 38.93 | 0.00 |
| SGR (%/d) | 6.54±0.01[a] | 5.33±0.02[b] | 55.36 | 0.00 |
| FI (g/fish d$^{-1}$) | 1.21±0.12[a] | 0.86±0.01[b] | 3.051 | 0.04 |
| FCR (FCR) | 0.99±0.15[a] | 1.24±0.01[b] | -2.97 | 0.04 |

Results are expressed as mean ± SE (standard error) (n = 3).

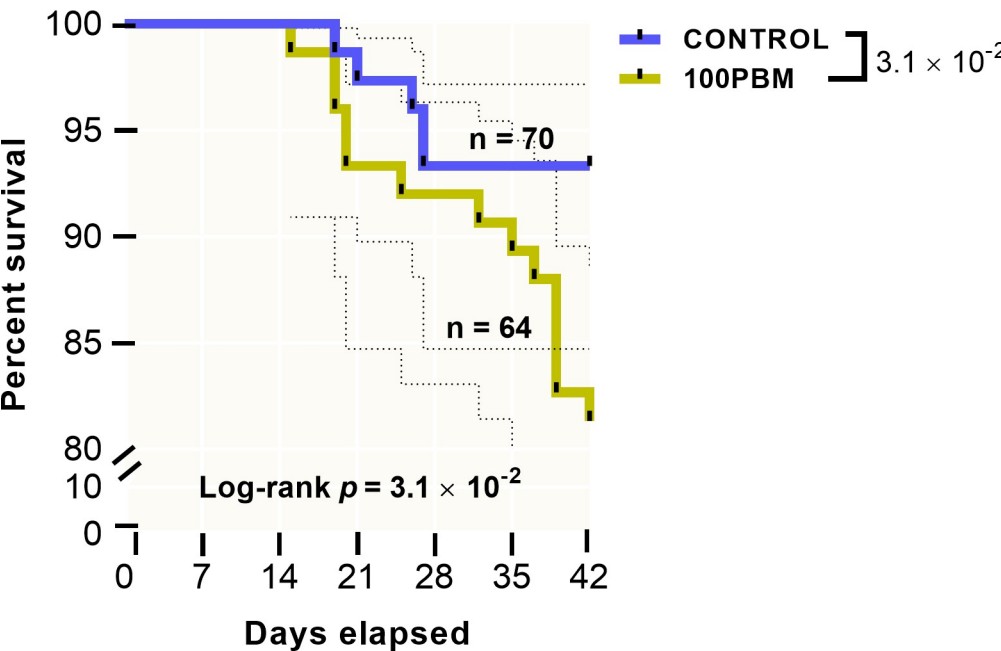

**Fig 1. Survival rate based on Kaplan-Meier survival analysis with Log-rank (Mantel-Cox) test of barramundi after 42 days feeding with either basal diet or PBM based diet.** Dotted line in survival plot indicates 95% confidence interval and *P* value indicate significant at 0.05.

## Intestinal morphology

The distal intestine of barramundi fed control (Fig 3A) and PBM (Fig 3B) were examined by transmission electron microscope. Microvilli height (Fig 3D) ($t = 6.727$, df = 28, $P < 0.0001$) and diameter (Fig 3E) ($t = 3.494$, df = 28, $P = 0.0016$) of barramundi fed PBM diet was significantly lower than barramundi fed control diet.

## Liver enzymes, immunity and stress related genes

Liver enzymes (AST and GLDH), immune response including serum lysozyme and bactericidal activity and stress related genes (HSP70 and HSP90) were significantly induced by the experimental diets (Fig 4). AST and GLDH in PBM fed fish was significantly higher than the control ($t = 2.268$, df = 10, $P = 0.047$ and $t = 3.199$, df = 10, $P = 0.010$) (Fig 4A and 4B), while serum lysozyme decreased significantly in PBM fed fish compared to control ($t = 2.842$, df = 10, $P = 0.018$) (Fig 4C). Meanwhile, none of the diets had significant effects on bactericidal activity ($t = 1.572$, df = 10, $P = 0.147$) (Fig 4D). In line with liver enzymes, similar results were observed in HSP70 and HSP90 when compared with control ($t = 2.905$, df = 10, $P = 0.016$ and $t = 5.102$, df = 10, $P = 0.001$) (Fig 4E and 4F).

## Antioxidant activity

Antioxidant activities of blood serum were significantly affected by total inclusion of PBM. Serum GPx activity declined significantly in PBM fed fish ($t = 2.833$, df = 10, $P = 0.017$) (Fig 5A), while MDA increased significant in PBM ($t = 2.251$, df = 10, $P = 0.048$) (Fig 5B) with respect to control.

**Table 6. Fatty acids (mg/100g on dry matter basis) of barramundi muscle when fed control and PBM based diet over a period of 42 days.**

| Fatty acid | Experimental diets | | Unpaired *t*-test | |
|---|---|---|---|---|
| | Control | 100PBM | *t*-value | *P*-value |
| C12:0 | 1.02±0.09[a] | 291.96±1.27[b] | -228.29 | 0.00 |
| C14:0 | 67.46±0.77[a] | 213.36±0.48[b] | -158.64 | 0.00 |
| C16:0 | 713.65±10.19[a] | 1253.24±68.93[b] | -7.75 | 0.00 |
| ΣSFA[1] | 1153.45±15.15 | 2320.11±65.41 | -17.38 | 0.00 |
| C16:1n7 | 130.78±1.90[a] | 286.88±2.21[b] | -53.82 | 0.00 |
| C20:1 | 36.62±0.38[a] | 111.68±2.97[b] | -25.07 | 0.00 |
| C14:1n5 | 0.97±0.03[a] | 6.30±0.06[b] | -84.46 | 0.00 |
| C18:1cis+trans | 859.76±5.73[a] | 2961.30±69.99[b] | -29.93 | 0.00 |
| ΣMUFA[2] | 1066.66±8.25[a] | 3412.55±74.01[b] | -31.50 | 0.00 |
| C18:3n3 | 55.12±0.38[a] | 213.57±3.10[b] | -50.94 | 0.00 |
| C20:5n3 (EPA) | 109.31±1.78 | 113.69±1.31 | -1.97 | 0.12 |
| C22:5n3 | 71.50±0.95[a] | 78.82±0.44[b] | -7.14 | 0.00 |
| C22:6n3 (DHA) | 683.13±12.43[a] | 370.84±1.99[b] | 24.82 | 0.00 |
| Σn-3 PUFA[3] | 940.67±15.93[a] | 821.03±11.05[b] | 5.24 | 0.01 |
| C20:3n6 | 25.68±0.57[a] | 49.16±2.84[b] | -8.07 | 0.00 |
| C20:4n6 | 92.29±1.90[a] | 114.06±3.54[b] | -5.43 | 0.01 |
| C18:3n6 | 17.51±1.42[a] | 51.44±6.52[b] | -5.08 | 0.01 |
| C22:4n6 | 62.67±1.07[a] | 23.06±0.26[b] | 36.20 | 0.00 |
| Σn-6 PUFA | 198.15±3.86[a] | 237.72±12.61[b] | -9.15 | 0.00 |
| Σn-3/Σn-6 | 4.75±0.04[a] | 3.47±0.18[b] | 6.78 | 0.00 |
| ΣPUFA[4] | 1485.98±24.56 | 1311.18±9.36 | 2.86 | 0.05 |
| Σn-3 LC-PUFA[3] | 868.55±15.25 | 569.45±2.78 | 19.29 | 0.00 |

Results are expressed as mean ± SE (standard error) (n = 3).

[1]Contains 10:0, 13:0, 15:0, 17:0, 18:0, 20:0, 21:0, 22:0 and 23:0.

[2]Contains C15:1, C17:1, C22:1n9 and C24:1.

[3]Contains C18:4n3, C20:3n3.

[4]Contains C18:2 trans, C18:2 cis, C20:2, C22:2.

Poultry by-product meal, PBM; saturated fatty acids, SFA; monounsaturated fatty acids, MUFA; polyunsaturated fatty acids, PUFA and LC-PUFA, long-chain polyunsaturated fatty acids (sum of 20:3n-3, 20:5n-3, 22:5n-3 and 22:6n-3).

Different superscripts letter indicate significant difference at *P* < 0.05, 0.01 and 0.001, followed by an unpaired student t-test.

## Discussion

A good number of studies have been devoted over the years to incorporate different levels of PBM, at the expense of FM in the diet of finfish and shellfish aquaculture [28] but most of the studies were performed on the basic nutritional aspects including proximate composition, amino acids, and fatty acid content, and its potential effect on the growth performance of the host fish [49–51]. In depth investigations are still lacking pertaining to the effects of PBM on the integrity of different organs, stress-related genes expression, or antioxidative responses in barramundi.

Establishing protein derived from animal industry as an ideal feed for finfish aquaculture, a series of studies have been conducted in Australia especially on barramundi. For instance, dietary inclusion of high-quality poultry protein concentrate from 5 to 20% demonstrated a reduced growth performance despite providing balanced amino acids in the diet but the reverse trend was observed when supplemented with phosphorous [8]. On the contrary, Siddik, Howieson [16] were able to replace 90% FM with either bioprocessed or unprocessed

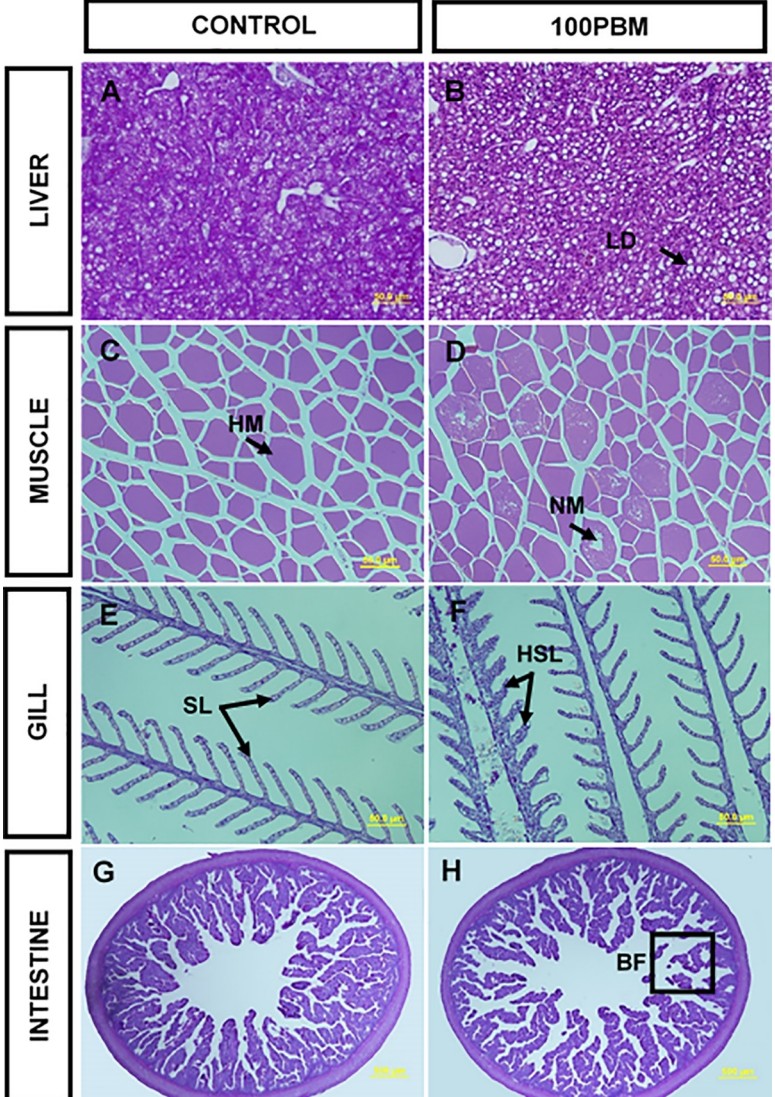

**Fig 2.** Liver (**A-B**), muscle (**C-D**), gill (**E-F**) (PAS stain; 40 × magnification; scale bar = 50 μm) and distal intestine (**G-H**) (PAS stain; 4 × magnification; scale bar = 500 μm) sections of barramundi fed control and PBM based diet at the end of 42 days of feeding trial. Lipid droplet, LD; healthy myotome, HM; necrotic myotome, NM; secondary lamellae, SL; hyperplasia in secondary gill lamellae, HSL and broken fold, BF.

PBM along with supplementation of fish protein hydrolysate with no apparent effects on the growth performance. In the present study, barramundi fed PBM supplemented with methionine affected the growth, feed utilization, FCR, and survival rate. Similarly, feeding barramundi non-FM based diet containing 450 g kg$^{-1}$ PBM and 285 g kg$^{-1}$ soybean meal impacted the growth, feed intake, and FCR despite supplementing taurine and the presumable reasons were palatability [29]. Deterioration in the growth performance of gibel carp, *Carassius auratus gibelio* was also observed when fed 100% animal protein containing PBM and meat and bone meal despite supplementation with methionine and lysine [52]. The possible reasons were the nutritional superiority or enhanced palatability in FM that could not be met up by PBM and MBM. Methionine, lysine and arginine levels in the 100PBM diet were at optimum level for barramundi growth [11] but histidine, isoleucine, and phenylalanine were lower

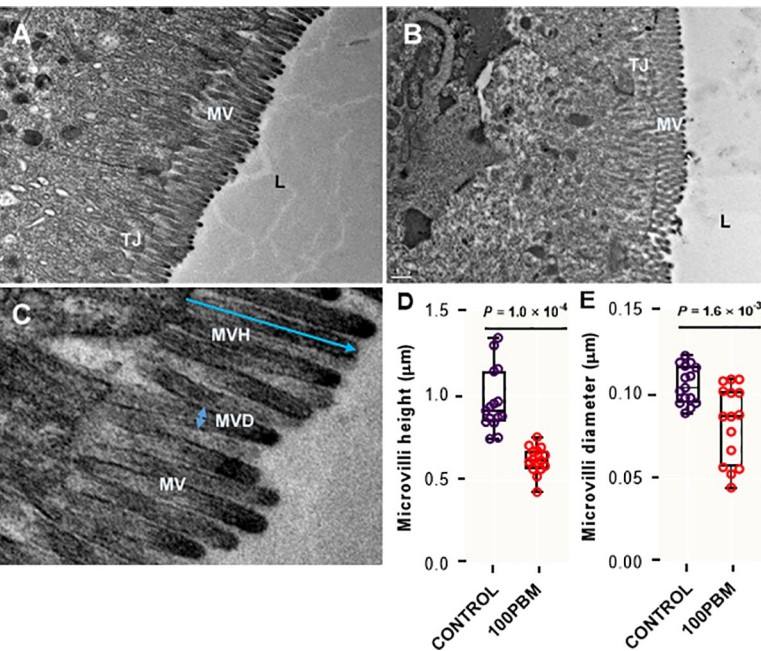

**Fig 3.** Observation of TEM in the intestine of barramundi fed Control (**A**) and PBM (**B**) at the end of 42 days of feeding trial. (**C**) Microvilli height and diameter measurement and comparison of microvilli height and diameter (panel **D & E**), performed by an unpaired student t-test at $P <0.05$ and 0.01. Microvilli, MV; Microvilli height, MVH; tight junction, TJ; microvilli diameter, MVD.

compare to FM based diet which may suppress the growth performance. Similarly, deficiency of histidine, methionine, isoleucine, lysine, and phenylalanine were identified to reduce the growth performance of spotted rose snapper, *Lutjanus guttatus* with higher inclusion of PBM [53]. Moreover, the abundance of MUFA and n-6 PUFA coupled with a deficiency of EFA particularly n-3 LC-PUFA, EPA and DHA in PBM were highlighted as one of the reasons for the reduced growth in Totoaba [54], catfish, *Ictalurus punctatus* [55], black sea turbot, *Psetta maeotica* [30] and gilthead sea bream, *Sparus aurata* L [56]. Similarly, higher MUFA content and lower levels of PUFA, in particular, n-3 LC-PUFA and EPA contents were found in 100PBM diets that could be responsible for the negative influence on growth, survival feed utilization and FCR. In addition, n-3 PUFA have been reported as an indispensable FAs for optimum growth and survival of many marine fish species [28, 56]. However, these findings contradict with the results of Panicz, Żochowska-Kujawska [57], Gunben, Senoo [58] and Shapawi, Ng [32] who reported no adverse effects of 100PBM on the growth and biometry indices of female tenches, *Tinca tinca*, tiger grouper juveniles, *Epinephelus fuscoguttatus and* humpback grouper, *Cromileptes altivelis*. This heterogeneity might be due to use different fish species and culture system or variability in nutritional composition palatability, and digestibility of PBM as it varies from batch to batch or among supplier companies [18, 28].

FAs composition of diet affects the FAs composition of fish muscle or meat which have been reported in many fish species [59–61]. In the present study, FAs of barramundi fillet were affected by the PBM diet. Total muscle SFA concentration was significantly higher in PBM fed fish may be due to the abundance of palmitic acid and myristic acid in fish muscle that are reflected in the PBM diet. A higher concentration of total SFA due to a high abundance of palmitic acid was observed in juvenile black sea bass fed 100PBM [62]. Muscle MUFA content in the present study increased in PBM fed fish which could be due to higher proportion MUFA

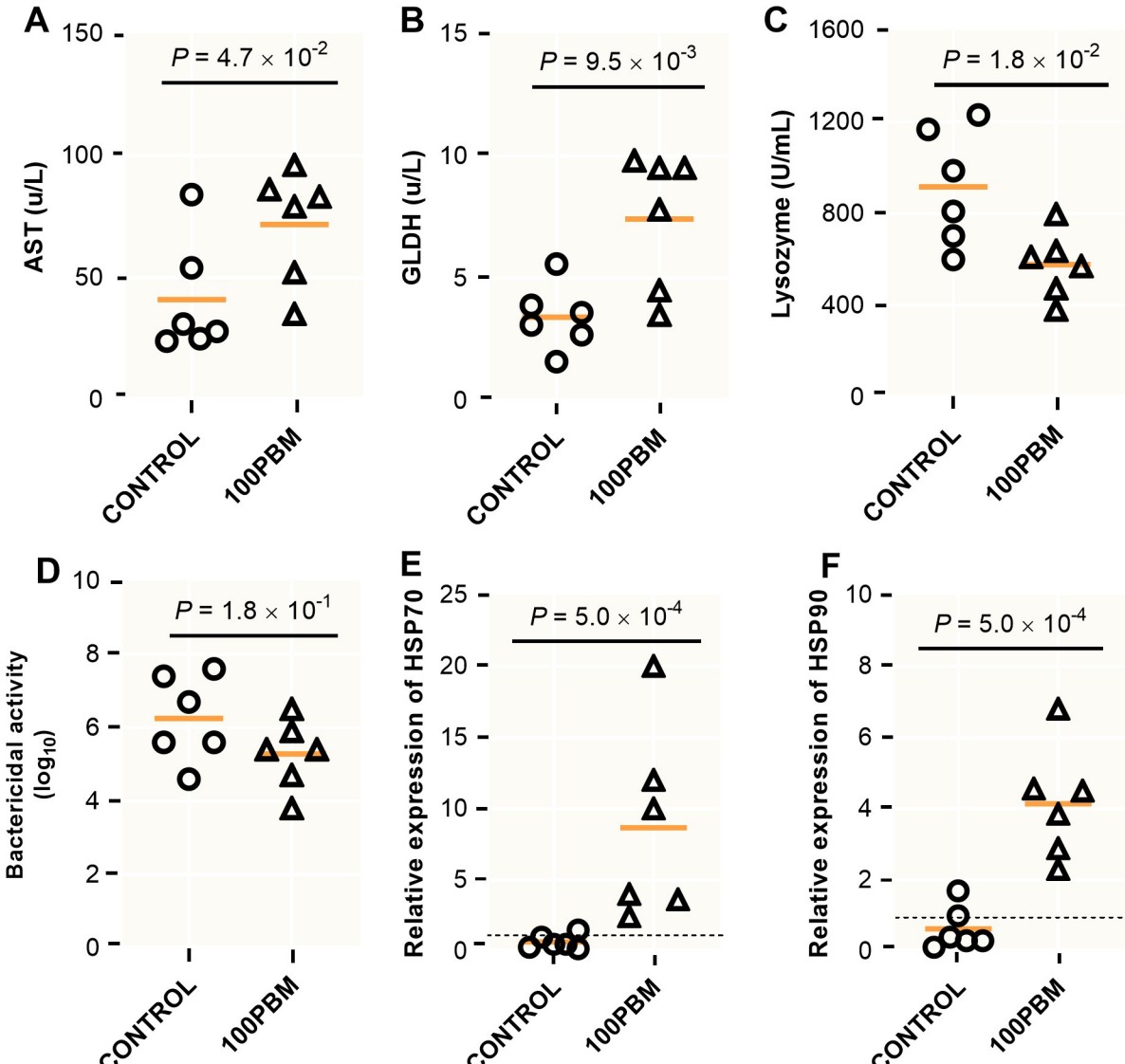

**Fig 4.** AST, aspartate aminotransferase (**A**) and GLDH, glutamate dehydrogenase (**B**), lysozyme (**C**), bactericidal activity (**D**) and heat shock related gene including HSP70 (**E**) and HSP90 (**F**) in barramundi after 42 days feeding with either control diet or PBM based diet. *P* values indicate significant at $P < 0.05$, 0.01 and 0.001, followed by an unpaired student *t*-test.

in the diet. This finding was similar to our earlier study [46]. Lower concentration of n-3 LC-PUFA and adrenic acid in fish muscle resulted in low total PUFA and n-3/n-6 ratio which are similar to the findings in barramundi fed high levels of PBM [7]. Similarly, 100PBM was lacking in essential fatty acids (EFAs) and also worsened the EFAs in the muscle of totoaba juveniles, *Totoaba macdonaldi* [54]. These results demonstrated that the total substitution of FM with PBM decreased PUFA levels in barramundi, which may consequently affect the nutritional value in terms of fatty acids available for human consumption.

It is well known that AST and GLDH are two important enzymes which primarily exist in liver at lower levels under normal condition but can leak into the blood rapidly when liver cells are damaged due to various stressors [63, 64]. In the present study, the PBM diet significantly increased the levels of AST and GLDH in the serum of barramundi, concomitant with the

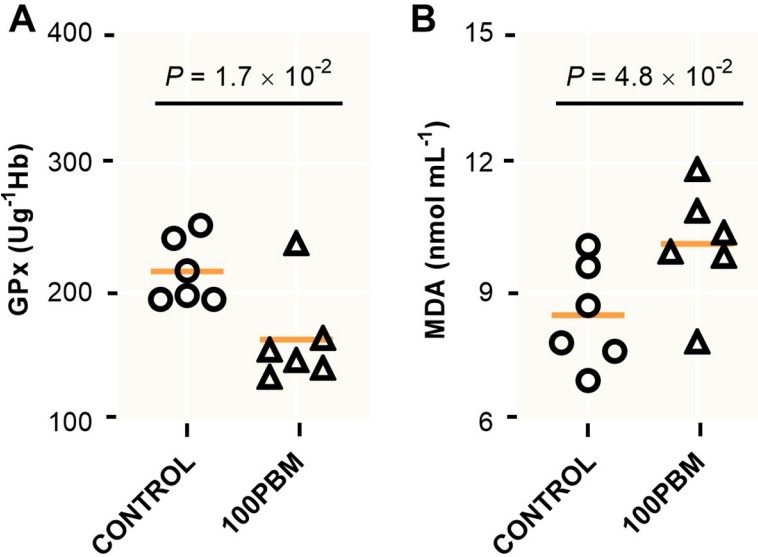

**Fig 5. (A)** GPx (Ug$^{-1}$ Hb) and **(B)** MDA (nmol mL$^{-1}$) in the serum of barramundi after 42 days feeding with either control diet or PBM based diet. *P* values indicate significant at *P* < 0.05, 0.01 and 0.001, followed by an unpaired student *t*-test.

histopathological damage of liver tissue. Likewise, plasma ALT was negatively impacted by the inclusion of animal protein blend (APB) (20% to 80%) in the diet of hybrid grouper while AST augmented significantly in 80% APB fed fish [38]. However, Panicz, Żochowska-Kujawska [57] reported no effects on blood biochemical parameters of juvenile tenches, *Tinca tinca* fed graded levels of PBM (25.7 to 100%).

To further clarify the effects of PBM on the liver function of barramundi, heat shock-related genes including HSP70 and HSP90 were examined. HSP70 and HSP90 are two important stress-related protein and their expression level elevate significantly when fish are exposed to different stressors, including pathogenic infection, crowding, poor water quality, and nutritionally deficient diet [65–67]. In the present study, both HSP70 and HSP90 upregulated significantly in the liver of barramundi that received 100PBM, indicating that 100% inclusion of PBM could act as a stressor.

In fish, immune functions of immune organs are strongly associated with the presence and activity of a unique array of molecules including lysozyme, complement proteins, immunoglobulins [68–70], and bactericidal activity that are influenced by the dietary modifications. Serum lysozyme was negatively triggered by PBM diet that support the findings of Subhadra, Lochmann [71, 72], who reported aggravated levels of complement and lysozyme activity in PBM fed largemouth bass, *Micropterus salmoides*.

Substitution of 100% FM with PBM resulted in lipidosis with clearly visible inflammation in the liver of juvenile tenches, *Tinca tinca* [57], supporting our present findings as hepatocyte lipid vacuolization with less amount of glycogen was observed in the liver tissue of barramundi fed PBM. The excessive amount of fat deposition in the liver negatively impacted the growth and immune response of fish [73] that are synchronous with the immunological results in the present study. Similarly, Siddik, Chungu [7] fed juvenile barramundi with different levels of PBM for 42 days and reported irregular liver arrangement with lipid deposition in the 100% PBM and bioprocessed PBM groups. Furthermore, higher administration of animal protein blend affected the morphology of the liver of hybrid grouper, characterized by hepatic vacuoles and a high amounts of lipid droplets which is a sign of hepatic steatosis [38]. The lipid

accumulation in the liver may occur when dietary lipid exceeds the capacity of the hepatic cells to oxidize which lead to synthesize and deposit larger amounts of triglyceride in vacuoles [38, 73, 74].

Muscle structure is the determinant of fish growth and can be affected by nutritionally-deprived diet [75]. For example, nutritional deficiency altered the muscle structure of Atlantic salmon, *Salmo salar* including myodegeneration [76]. Similarly, fish fed PBM diet showed necrosis and fibre degeneration in muscle. Gill is one of the important immune organs in fish and its structure can be affected by stress and diet [77]. In the present study, hyperplasia in secondary gill lamellae was in PBM fed fish but the possible reasons are not well understood, deserving further study.

Evaluating intestinal morphology in response to dietary changes is important to determine the health status and welfare of fish. Intestinal morphology, in particular, villous structure, and microvillus height and diameter is related to absorption and assimilation of nutrient and immunological function [16, 78, 79]. Histological analysis showed that broken and short fold in the present study in PBM fed groups are in line with TEM results, showing significantly smaller with a shorter diameter of microvilli, which are responsible for the lower efficiency of nutrient uptake, thus suppressing the growth and survival. Similar results were reported by Siddik, Howieson [16] who found significantly lower microvilli height in the distal intestine of barramundi after 56 days post-feeding with 10% supplemented 90PBM. Hence total replacement of FM with PBM impacted the welfare of barramundi, as reflected by the histological and TEM analysis.

Antioxidant status in fish, as determined by several antioxidant enzymes including CAT, SOD, GPx, and MDA have been considered as the first line of defensive biomarkers to protect cells and tissues from oxidative damage, caused by some free radicals such as superoxide anion ($O^{2-}$), hydrogen peroxide ($H_2O_2$) and hydroxyl radical (OH) [80, 81]. Glutathione peroxidase, GPx is an important antioxidant enzyme showing strong radical-scavenging capacity against free radicals and lipid peroxides [82]. The present study detected a significantly lower activity of GPx in the serum of barramundi fed with PBM, which may due to the lower levels of n-3 LC-PUFA and DHA in 100PBM diet. Liu, Mai [83] reported that marine fish are susceptible to oxidative stress due to their high demand for LC-PUFA. However, elevated serum GPx activity in barramundi fed 90% fermented PBM supplemented with tuna hydrolysate [17] might be due to the antioxidant capacity of fish protein hydrolysate [84]. It has been reported that the GPx activity is well correlated with the concentration of MDA [85]. MDA is a natural biomarker and main ending product of lipid peroxidation [86, 87] and its elevation indicates oxidative injury [88] and associates with the pathological state of animals including cell structure damage and function [86, 87]. A lower activity of GPx with the higher level of MDA indicates that PBM based diet may provoke the oxidative damage of barramundi which was further proven by the presence of hepatocyte lipid vacuolization.

In summary, regardless of methionine supplementation, the total replacement of FM with PBM is not nutritionally adequate for barramundi, as indicated by depressed growth performance and immune response. An unfavourable effect of a PBM based diet was observed on antioxidant enzymes. Also, adding PBM induced the lipid droplet in the liver for barramundi via affecting the expression levels of heat shock related genes and liver enzymes. Feeding PBM not only triggered the fiber degeneration and necrosis in muscle and hyperplasia in gills but also induced the intestinal villus morphology by decreasing intestinal microvilli morphology, which may suggest that high levels of PBM could impair the welfare of barramundi. Further studies need to be conducted along with supplementation of other EAA and/or EFA with PBM to investigate the welfare of farmed barramundi.

## Acknowledgments

The authors are also thankful to the Australian Centre for Applied Aquaculture Research (ACAAR), Fremantle, Australia for providing fish and Rowan Kleindienst for technical assistance during fish husbandry. We are also sincerely thankful to Dr. Fran Stephens for helping to collect histological samples and prepare histological slides.

## Author Contributions

**Conceptualization:** Md. Reaz Chaklader, Ravi Fotedar.

**Formal analysis:** Md. Reaz Chaklader, Muhammad A. B. Siddik.

**Funding acquisition:** Ravi Fotedar.

**Investigation:** Md. Reaz Chaklader.

**Supervision:** Ravi Fotedar.

**Validation:** Ravi Fotedar.

**Writing – original draft:** Md. Reaz Chaklader.

**Writing – review & editing:** Muhammad A. B. Siddik, Ravi Fotedar.

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
