## [Decision Letter · Decision Letter 0]

30 Apr 2020

PONE-D-20-07405

Total replacement of fishmeal with poultry by-product meal affected the growth, muscle quality, histological structure, antioxidant capacity and immune response of juvenile barramundi, Lates calcarifer

PLOS ONE

Dear Mr Chaklader,

Thank you for submitting your manuscript to PLOS ONE. After careful consideration, we feel that it has merit but does not fully meet PLOS ONE’s publication criteria as it currently stands. Therefore, we invite you to submit a revised version of the manuscript that addresses the points raised during the review process.

We would appreciate receiving your revised manuscript by Jun 14 2020 11:59PM. To enhance the reproducibility of your results, we recommend that if applicable you deposit your laboratory protocols in protocols.io, where a protocol can be assigned its own identifier (DOI) such that it can be cited independently in the future. For instructions see: http://journals.plos.org/plosone/s/submission-guidelines#loc-laboratory-protocols

We look forward to receiving your revised manuscript.

Kind regards,

Juan J Loor

Academic Editor

PLOS ONE

2. We note you have included a table to which you do not refer in the text of your manuscript. Please ensure that you refer to Table 1 in your text; if accepted, production will need this reference to link the reader to the Table.

Reviewers' comments:

Reviewer's Responses to Questions

**Comments to the Author**

1. Is the manuscript technically sound, and do the data support the conclusions?

Reviewer #1: Partly

2. Has the statistical analysis been performed appropriately and rigorously? 

Reviewer #1: Yes

3. Have the authors made all data underlying the findings in their manuscript fully available?

Reviewer #1: Yes

4. Is the manuscript presented in an intelligible fashion and written in standard English?

Reviewer #1: No

5. Review Comments to the Author

Reviewer #1: The authors present an interesting and well-constructed experiment evaluating fishmeal and poultry meal based diets fed to juvenile barramundi. There are consistent issues throughout with the writing style not being in a scientific contest, the description of results, discussion and conclusions need to be improved. Quiet clear is the lack of appropriate citation relevant to the study.

6. PLOS authors have the option to publish the peer review history of their article (what does this mean?). If published, this will include your full peer review and any attached files.

Reviewer #1: No

---

## [Author Response · Author response to Decision Letter 0]

17 Jul 2020

RESPONSE TO REVIEWER

General comments: 

The authors present an interesting and well-constructed manuscript evaluating fishmeal and poultry meal based diets fed to juvenile barramundi. There are consistent issues throughout with the writing style in a scientific contest, the description of results, discussion and conclusions need to be improved. Quiet clear is the lack of appropriate citation and relevance to the study. My most obvious concern with the work is the lack of understanding in the field, in particular with the species in question. The barramundi is a very well understood species, with a lot of very old publication and work undertaken by various labs predominantly in Australian and south east Asia. This dates back around 30 years. I can find specific reference to the use of poultry meal in barramundi diets published in 1998. So, this is by no means a novel raw material for barramundi. The use of terrestrial meals as protein in barramundi is fundamentally well understood, and taking a deep dive we can see that there are studies that even evaluate multiple poultry meal types and their level of processing on the performance of barramundi. So this is not a new area of research. Moreover, their selection of relevant references is odd given there are numerous papers specifically on barramundi yet they chose to cite authors working on vastly different fish in the introduction. Its just a comment rather than criticism but I think should be revised.

Response: We have revised introduction and discussion with relevant citations. We fully acknowledge the fact that the PBM is not a novel raw material used in barramundi and their inclusion in the diet of aquaculture species is not a new area of research. To date, many of the studies, have focused on the nutritional value of PBM, including proximate composition, amino acids and fatty acids contents and its effect on fish growth. Though, research on the utilization of PBM started 30 years before, still there is no consistency in the results when PBM is fed to barramundi or other carnivorous fish species. The results, so far, have been mixed in terms of effective quantities of dietary inclusion of PBM, relationship between inclusion levels with growth performance of the host species, including barramundi, nutritional profiles of the PBM included diets, etc. However, there surely exists lack of information and the understanding on the effects of PBM on the architecture of different organs or other physiological parameters including stress related oxidative biomarkers and stress related genes. A recent studies in our laboratory was able to replace 90% of fishmeal with PBM assisted with the supplementation of fish protein hydrolysate (Siddik et al., 2019) while other study was able to include only 20% of fishmeal, with high quality poultry protein along with phosphorous supplementation (Simon et al., 2019). In addition, a recent study of Ye et al., (2019) reported that feeding graded levels of animal protein blend (20-80%) containing poultry by-product meal, shrimp meal and spray-dried blood meal negatively impacted the growth, feed efficiency and protein efficiency ratio, nutrient composition of lipid, liver structure and plasma biochemical parameters of juvenile hybrid grouper (Epinephelus fuscoguttatus♀× Epinephelus lanceolatus♂). Hence, this inconsistence in previous studies reinforced with molecular data have motivated us to conduct this research in order to improve the utilization of PBM in barramundi diet. The outcome of this research being the current MS was aimed to replace one hundred percent of fishmeal with PBM not only using growth but other physiological and molecular approaches. 

Siddik MAB, Howieson J, Fotedar R. Beneficial effects of tuna hydrolysate in poultry by-product meal diets on growth, immune response, intestinal health and disease resistance to Vibrio harveyi in juvenile barramundi, Lates calcarifer. Fish and Shellfish Immunology. 2019;89:61–70.

Simon CJ, Salini MJ, Irvin S, Blyth D, Bourne N, Smullen R. The effect of poultry protein concentrate and phosphorus supplementation on growth, digestibility and nutrient retention efficiency in barramundi Lates calcarifer. Aquaculture. 2019;498:305-14.

Ye H, Zhou Y, Su N, Wang A, Tan X, Sun Z, et al. Effects of replacing fish meal with rendered animal protein blend on growth performance, hepatic steatosis and immune status in hybrid grouper (Epinephelus fuscoguttatus♀ × Epinephelus lanceolatus♂). Aquaculture. 2019;511:734203.

Introduction

The introduction is written in a way that requires attention. It is going all over the place from rendered animal meals, unusual reference to unrelated obscure species, LC-PUFAs, lysine content, growth hormones, methionine metabolism, health status (immune, antioxidant), aquatic pollution. This distracts from the main purpose and history of R&D on barramundi in Australia and elsewhere. 

Reponses: we have completely revised the introduction. Please see lines 65-132.

Methods

• The Diet formulas are presented in table 1, however the title indicates the test diets were supplemented with protein hydrolysates. This is clearly from another study and cut and paste. 

Response: We are sorry for unintentional mistake. This sentence was included here by mistake as we were completing our MS which has been now published. We have corrected the Table 1 now.

• The Fatty Acid analysis section lacks a lot of detail. 

Response: We have updated the methodology of fatty acids analysis. Please see lines 223-230.

• Ive never seen fatty acids and amino acids presented side by side like that. Its odd.

Response: We have separated the fatty acids and amino acids in different tables now. We have also added amino acids profile of PBM in Table 3. 

• Antioxidant status assessment lacks a lot of detail

Response: We have revised and please see line 249-252. 

• In the calculation, there are 2 things requiring attention, AI and TI. 

Response: We have deleted AI and TI from all text.

Results

• You really need a table of results here that clearly show the values +/- SE for your data. The figure 1 is not satisfactory. FBG acronym not explained. Fig 1 appears to have individual fish (n=64) in some and tank replication in the other (n=3). Not clear as to why this is done differently. 

Response: We have deleted Figure 1 and have presented growth related data in Table 5 adding mean ± SE. 

• Muscle FA section poorly written. 

Response: we have elaborated and re-written the muscle FA section. Please see lines 336-342.

• Table 3. Units? Also, check stats. Row C18:1cis+trans are very different values whereas C17:0 are almost the same (44 v 40) yet highly significant. 

Response: We have mentioned units in Table 2 and Table 6, and also double checked the statistical analysis.

• Table 3. What have AI and TI got to do with the FAME? 

Response: We have deleted AI and TI from all over the text.

• In Histomorphology no mention of the gill and intestine result (Fig 2)

Response: Corrected. Please see line 364-367.

Discussion 

• A few studies? There are dozens specifically investigating this material in fish and prawns. 

Response: Agreed. We have rewritten the whole paragraph. Please see lines 477-517. 

• Have they already done this study ref 32? 

Response: No, we have not done this. 

• In terms of Methionine supplementation, the authors ended up with different levels of Met in the diets despite their attempts to balance. Either way the inclusion in both diets is well above the known requirement for the species and I doubt this would be a relevant reason for the differences observed. Need to discuss the results in more detail with a focus on the things that are explainable. 

Response: According to the Millamena (1994) and Glencross (2006), the requirements for methionine, lysine and arginine was reported to be about 22, 49 and 38 g kg-1 of dietary protein, respectively. We have not added methionine in the control diet since the control diet has a sufficient level of methionine for barramundi normal growth (Table 3). But methionine was lacking in PBM in the present study (Table 3) and we added 0.40% in the test diet to meet the required level for barramundi growth. However, we have thoroughly revised the discussion. Please see lines 497-511.

Millamena, O.M. (1994) Review of SEAFDEC/AQD fish nutrition and feed development research. In:Feeds for Small-Scale Aquaculture, Proceedings of the National Seminar-Workshop on Fish Nutrition and Feeds(Santiago, C.B., Coloso, R.M., Millamena, O.M. & Borlongan, I.G. eds), pp. 52–63. SEAFDEC Aquaculture Department, Iloilo, Philippines.

Glencross B. The nutritional management of barramundi, Lates calcarifer – a review. Aquaculture Nutrition. 2006;12(4):291-309.

• Similarly, with EFA story there are reports on fatty acids nutrition specifically in barramundi but these have largely been ignored in favour of other more obscure species. Moreover, this study is primarily an evaluation of a protein rich ingredient to replace fishmeal, so we are crossing over a lot here with lipid specific studies.

Response: we have thoroughly revised the fatty acids section in discussion part with relevant citations particularly on barramundi and other closely related species. Please see lines 519-531.

• Generally, the authors are describing their methods here but not relating it to or explaining in light of the present study. Take the CAT, SOD, GPX section for example.

Response: We have thoroughly revised the paragraph with relevant citation and underlying reasons behind the influence of antioxidant enzymes. Please see the line 586-601.

Acknowlegements

• Its Dr Fran Stephens. This clearly indicates your lack of understanding and probably had a lot of help. Fran is an expert in her field and should at least have her name correctly acknowledged. If she has helped in this analysis and interpretation of data, maybe consider her as a co-author?

Response: We are extremely sorry for making mistake to write Dr Fran Stephens’s name. Please see line 617. All histological analysis and data interpretation have done by authors. Only we have outsourced work from her related to histological slides only.

---

## [Decision Letter · Decision Letter 1]

9 Oct 2020

PONE-D-20-07405R1

Total replacement of fishmeal with poultry by-product meal affected the growth, muscle quality, histological structure, antioxidant capacity and immune response of juvenile barramundi, Lates calcarifer

PLOS ONE

Dear Dr. Chaklader,

Thank you for submitting your manuscript to PLOS ONE. After careful consideration, we feel that it has merit but does not fully meet PLOS ONE’s publication criteria as it currently stands. Therefore, we invite you to submit a revised version of the manuscript that addresses the points raised during the review process.

We look forward to receiving your revised manuscript.

Kind regards,

Mahmoud A.O. Dawood, PhD

Academic Editor

PLOS ONE

Additional Editor Comments (if provided):

The authors are encouraged to revise the manuscript in the light of reviewer 3 comments.

Reviewers' comments:

Reviewer's Responses to Questions

**Comments to the Author**

1. If the authors have adequately addressed your comments raised in a previous round of review and you feel that this manuscript is now acceptable for publication, you may indicate that here to bypass the “Comments to the Author” section, enter your conflict of interest statement in the “Confidential to Editor” section, and submit your "Accept" recommendation.

Reviewer #1: (No Response)

Reviewer #2: All comments have been addressed

Reviewer #3: (No Response)

2. Is the manuscript technically sound, and do the data support the conclusions?

Reviewer #1: Yes

Reviewer #2: Yes

Reviewer #3: (No Response)

3. Has the statistical analysis been performed appropriately and rigorously? 

Reviewer #1: Yes

Reviewer #2: Yes

Reviewer #3: (No Response)

4. Have the authors made all data underlying the findings in their manuscript fully available?

Reviewer #1: Yes

Reviewer #2: Yes

Reviewer #3: (No Response)

5. Is the manuscript presented in an intelligible fashion and written in standard English?

Reviewer #1: Yes

Reviewer #2: Yes

Reviewer #3: (No Response)

6. Review Comments to the Author

Reviewer #1: This revision is greatly improved. Well done. If i had to be picky i would have said that the methionine requirement in barramundi was revised a few years back and there are a couple of references on the subject, so the older reference #8 cited is actually quite outdated. Having said that the MS reads well and is ready for publication.

Reviewer #2: (No Response)

Reviewer #3: The Manuscript is well designed and conducted and presents an interesting subject to the marine aquaculture industry. However, it did not tell the optimum inclusion level of poultry by product meal with and without methionine supplementation, which is a very important issue for the journal readers and for application.

Minor comments

The proximate composition of diets lacks information on the mineral contents and energy values. Also, all values in Tables 1-3 miss +/- SE (SD)

7. PLOS authors have the option to publish the peer review history of their article (what does this mean?). If published, this will include your full peer review and any attached files.

Reviewer #1: No

Reviewer #2: **Yes: **Dr. Md. Tawheed Hasan

Reviewer #3: **Yes: **Mabrouk ELSABAGH

---

## [Author Response · Author response to Decision Letter 1]

21 Oct 2020

Response to Reviewer 1

Reviewer #1: This revision is greatly improved. Well done. If i had to be picky i would have said that the methionine requirement in barramundi was revised a few years back and there are a couple of references on the subject, so the older reference #8 cited is actually quite outdated. Having said that the MS reads well and is ready for publication.

Response: We have added updated reference. Please see line 151.

Response to Reviewer 3

Reviewer #3: The Manuscript is well designed and conducted and presents an interesting subject to the marine aquaculture industry. However, it did not tell the optimum inclusion level of poultry by product meal with and without methionine supplementation, which is a very important issue for the journal readers and for application.

Minor comments

The proximate composition of diets lacks information on the mineral contents and energy values. Also, all values in Tables 1-3 miss +/- SE (SD)

Response: 

Thank you for your complements regarding the design of the experiment and its conduction. Our published earlier article (Chaklader et al., 2020) has proved that 90% poultry by product meal along with the supplementation of fish protein hydrolysate can replace the fishmeal with some positive physiological outcomes for barramundi. As we found that methionine was lacking in the poultry by product meal for enhancing the growth of barramundi and therefore the current study was aimed to investigate if methionine supplementation alone in poultry by product meal could replace fishmeal entirely from the diet of barramundi. However, the current results clearly indicated that methionine alone is not able to replace the contribution of fish protein hydrolysate in barramundi diet. As fish protein hydrolysate have other nutrient ingredients (such as biologically active low molecular weight peptides) those in combination with poultry by product meal are able to replace fishmeal entirely in barramundi diets.

The results of previously published article and current article under review clearly indicate although methionine is essential for barramundi’s physiological performance, fish hydrolysate in addition to methionine has other beneficial micro nutrient embedded in it. Hence, optimum level of poultry by product meal in barramundi diet is 90% of the protein source along with fish hydrolysate without any addition of pure methionine. The optimum inclusion level of poultry by product meal with purified methionine supplementation is yet to be investigated.

Chaklader MR, Fotedar R, Howieson J, Siddik MAB, Foysal MJ. 2020. The ameliorative effects of various fish protein hydrolysates in poultry by-product meal diets on muscle quality, serum biochemistry and immunity in juvenile barramundi, Lates calcarifer. Fish & Shellfish Immunology, 104:567-578. https://doi.org/10.1016/j.fsi.2020.06.014

Proximate composition has been revised (Table 1). Table 1 is the calculation of feed formulation with analysed proximate composition while amino acid (Table 2) and fatty acid composition (Table 3) was analysed for one sample repeated in triplicate. One sample/treatment particularly for diets is accepted in aquaculture study as it is depicted in various publication (Siddik et al., 2019, Chaklader 2020, Ye et al., 2019 and Sabbagh et al., 2020). 

Siddik MAB, Chungu P, Fotedar R, Howieson J (2019) Bioprocessed poultry byproduct meals on growth, gut health and fatty acid synthesis of juvenile barramundi, Lates calcarifer (Bloch). PLoS ONE 14(4): e0215025. https://doi. org/10.1371/journal.pone.0215025

Chaklader MR, Fotedar R, Howieson J, Siddik MAB, Foysal MJ. 2020. The ameliorative effects of various fish protein hydrolysates in poultry by-product meal diets on muscle quality, serum biochemistry and immunity in juvenile barramundi, Lates calcarifer. Fish & Shellfish Immunology, 104:567-578. https://doi.org/10.1016/j.fsi.2020.06.014

Ye, H., Zhou, Y., Su, N., Wang, A., Tan, X., Sun, Z., Zou, C., Liu, Q., Ye, C., 2019. Effects of replacing fish meal with rendered animal protein blend on growth performance, hepatic steatosis and immune status in hybrid grouper (Epinephelus fuscoguttatus♀ × Epinephelus lanceolatus♂). Aquaculture 511, 734203. 10.1016/j.aquaculture.2019.734203.

Sabbagh, M., Schiavone, R., Brizzi, G., Sicuro, B., Zilli, L., Vilella, S., 2019. Poultry by-product meal as an alternative to fish meal in the juvenile gilthead seabream (Sparus aurata) diet. Aquaculture 511, 734220.

---

## [Editor Report · Decision Letter 2]

27 Oct 2020

Total replacement of fishmeal with poultry by-product meal affected the growth, muscle quality, histological structure, antioxidant capacity and immune response of juvenile barramundi, Lates calcarifer

PONE-D-20-07405R2

Dear Dr. Chaklader,

We’re pleased to inform you that your manuscript has been judged scientifically suitable for publication and will be formally accepted for publication once it meets all outstanding technical requirements.

Kind regards,

Mahmoud A.O. Dawood, PhD

Academic Editor

PLOS ONE
---

## [Editor Report · Acceptance letter]

3 Nov 2020

PONE-D-20-07405R2 

Total replacement of fishmeal with poultry by-product meal affected the growth, muscle quality, histological structure, antioxidant capacity and immune response of juvenile barramundi, *Lates calcarifer*

Dear Dr. Chaklader:

I'm pleased to inform you that your manuscript has been deemed suitable for publication in PLOS ONE. Congratulations! Your manuscript is now with our production department. 

Kind regards, 

on behalf of

Dr. Mahmoud A.O. Dawood 

Academic Editor

PLOS ONE